# DB-GPT-Hub: Towards Open Benchmarking Text-to-SQL Empowered by Large Language Models

## Abstract

Large language models (LLMs) becomes the dominant paradigm for the challenging task of text-to-SQL. LLM-empowered text-to-SQL methods are typically categorized into prompting-based and tuning approaches. Compared to prompting-based methods, benchmarking fine-tuned LLMs for text-to-SQL is important yet under-explored, partially attributed to the prohibitively high computational cost. In this paper, we present *DB-GPT-Hub*, an open benchmark suite for LLM-empowered text-to-SQL, which primarily focuses on tuning LLMs at large scales. The proposed benchmark consists of: 1. a standardized and comprehensive evaluation of text-to-SQL tasks by fine-tuning medium to large-sized open LLMs; 2. a modularized and easy-to-extend codebase with mainstream LLMs and experimental scenarios supported, which prioritizes fine-tuning methods but can be easily extended to prompt-based setting. Our work investigates the potential gains and the performance boundaries of tuning approaches, compared to prompting approaches and explores optimal solutions tailored to specific scenarios. We hope *DB-GPT-Hub*, along with these findings, enables further research and broad applications that would otherwise be difficult owing to the absence of a dedicated open benchmark. The project code has been released anonymously at https://github.com/anonymity-360/DB-GPT-Hub.

## 1 Introduction

The task of text-to-SQL, which converts natural utterances into SQL queries, has emerged as a popular topic in both natural language processing and database (Yu et al., 2018b; Deng et al., 2022). It effectively narrows the gap between non-expert users and database systems, significantly enhancing data processing efficiency. Essentially, text-to-SQL can be characterized as a sequence-to-sequence modeling task (Sutskever et al., 2014), where the database schema and the natural language question are transformed into a sequential input, while the desired SQL query serves as the sequential output target. Early works focus on fine-tuning domain-specific Transformer models and developing decoding techniques specifically for the task, leveraging SQL syntax, semantics, and the complex interplay between questions and databases (Scholak et al., 2021; Qi et al., 2022).

While recently large language models (LLMs) such as ChatGPT (Brown et al., 2020) and GPT-4 (OpenAI, 2023a) have showcased their remarkable capabilities in engaging in human-like communication and understanding complex queries, LLMs have emerged as a new paradigm for text-to-SQL (Liu et al., 2023; Trummer, 2022). Notably, since 2023, the majority of top-performing solutions on the Spider leaderboard (Yale, 2018) have been methods based on LLMs.

The most recent advancement in this area involves employing LLMs for generating accurate SQL queries through in-context learning (ICL) techniques, notably zero-shot and few-shot prompting (OpenAI, 2023b; Dong et al., 2023; Pourreza & Rafiei, 2023). Beyond the inherent challenge of ambiguity and complexity, the laborious efforts for annotating SQL query-response exemplars by domain experts hinder the process of scaling-up data hungry LLMs for text-to-SQL applications. Meanwhile, another prominent approach is fine-tuning LLMs using additional task-specific training data to enhance their efficacy for text-to-SQL tasks  Li et al. (2023a); Sun et al. (2023). The remarkable performances achieved in these works indicate the immense potential of fine-tuning. However, compared to prompting approaches, fine-tuning approaches have been relatively under-explored, partially attributed to the prohibitively high computational cost. Recent systematic studies

(Gao et al., 2023; Zhang et al., 2024) still mainly highlight the ICL abilities of LLMs and their accuracy in generating SQL queries in relevant tasks.

Up until now, there still has not been a universally acknowledged open benchmark for tuning approaches, which impedes researchers and practitioners from comparing methods and reproducing results, potentially slowing down advancement in this field. As a first step towards addressing these challenges, in this work, we present a holistic framework, namely *DB-GPT-Hub*,. **Apart from existing works that mostly focus on few-shot prompting strategies or tuning relatively smaller LLMs, our work focuses on tuning larger LLMs**. In all, DB-GPT-Hub consolidates essential research assets (e.g., data, model services, evaluation methods, documentation) with following distinct merits:

- **Standardization**. We establish a standardized pipeline in an open-source codebase, with unified experimental settings and containerized environments, to enable transparent and consistent comparisons of LLM models after text-to-SQL tasks tuning.

- **Comprehensiveness**. We conduct extensive benchmarking that covers a range of medium to large-sized, fine-tuned LLMs across various experimental scenarios and explore their relative performance compared to prompting methods. Our work comprises one of the most pragmatic and expansive sets of benchmark suites available.

- **Extensibility**. As a rapidly evolving field, novel LLM-based methods constantly emerge, and the best practice continuously evolves. Following our documentation and protocols, one could effortlessly incorporate novel ingredients into our codebase: new datasets, new modules, new models (or model services), and new evaluation programs. Moreover, our framework offers easy compatibility with various prompting techniques. The high extensibility will eventually benefit the research area of text-to-SQL.

## 2 BACKGROUND AND PROBLEM FORMULATION

### 2.1 A GENERALIZED SETUP

The input of text-to-SQL task is a natural language question $q$ and the database information $\mathcal{D}$. The output is the SQL $s$ corresponding to the question. The database $\mathcal{D} = \{S, K_p, K_f\}$ includes database schema $S$, primary keys $K_p$ and foreign keys $K_f$, where $S$ usually contains multiple tables $T_k : S = \{T_1, T_2, ...T_s...\}$. Each table $T_k$ has table name $N_k$, column names $c_j$ and column data types $t_j$. Therefore, $T_k = \{N_k, (c_{k1}, t_{k1}), (c_{k2}, t_{k2})...\}$. Consider the queries may come from various database domains, we formulate the data into a set of triples $\mathcal{M} = \{(q_i, s_i, \mathcal{D}_i)\}$, with $i$ denoting the index of the query, the output and source database.

### 2.2 PROMPT-BASED AND FINE-TUNING SETTINGS

Based on how LLMs are used for text-to-SQL generations, the problem settings can be categorized into two scenarios: zero-shot/few-shot prompting and fine-tuning.

**Zero-shot / Few-shot Prompting.** In zero-shot scenarios, no exemplar is provided while in few-shot a few input–output exemplars are provided to prompt LLMs. Formally, given a pretrained LLM parameterized by $\theta$, the question $q_i$, and $k$ exemplars ($k \geq 0$), the objective is maximize the probability of generating the correct SQL $s_i$ from the LLM:

$$\max_{s_i} \mathbb{P}_{LLM_\theta}(s_i | \sigma(q_i, \mathcal{M})), \quad |\mathcal{M}| = k \tag{1}$$

where $\Theta$ and $\sigma(q_i, \mathcal{M})$ [1] denotes a representation space of the target question $q_i$ by incorporating relevant information from exemplars.

**Fine-tuning.** The fine-tuing process involves adapting the pretrained $LLM_\theta$ to generate SQL from the input sequences by tuning the model with text-to-SQL datasets, which contain a collection of serialized inputs $q_i$ and corresponding SQL outputs $s_i$ pairs. The object of fine-tuning is minimize

---

[1] $\sigma(q_i, \mathcal{M})$ technically denotes the information set generated by $q_i$ and $\mathcal{M}$.

the empirical loss:

$$\min_{\theta} \mathcal{L}(\widehat{s}_i(LLM_\theta), s_i|\sigma(q_i)), \tag{2}$$

where $\mathcal{L}$ is the loss function to measure the difference between the generated SQL and the groundtruth.

Despite the significant advances achieved with few-shot prompting of LLMs, it remains a formidable challenge for a pretrained LLM to rely solely on its parametric knowledge and prompting to accurately process highly complex SQL queries.

**Parameter-Efficient Fine-tuning.** Medium to large-sized models with billions of parameters, are prohibitively expensive to fine-tune in order to adapt them to particular tasks or domains. Parameter-Efficient Fine-Tuning (PEFT) methods enable efficient adaptation of large pretrained models to various downstream applications by only fine-tuning a small number of (extra) model parameters instead of all the model's parameters. Two mostly commonly used techniques are LoRA (Hu et al., 2021), which proposes to freeze pretrained model weights and inject trainable layers (rank-decomposition matrices) in each transformer block, and its quantized version QLoRA (Dettmers et al., 2023). Throughout the benchmark, we use these two strategies consistently to tune the LLMs. See Section 3 and Section 4 for details of tuning benchmark design and experimental results.

## 3 BENCHMARK DESIGN AND RESOURCES

### 3.1 SETUP

**Datasets.** We conduct experiments mainly on the following 2 well recognized public datasets:

- Spider (Yu et al., 2018b). Spider is a large-scale cross-domain dataset consisting of 10,181 natural language queries, 5,693 unique complex SQL queries across 200 databases, covering 138 domains. The standard protocol for this dataset divides it into 8,659 training examples and a holdout of 2,147 test examples across 34 databases. SQL queries are categorized into four difficulty levels, i.e., easy, medium, hard and extra hard.
- BIRD (Li et al., 2023b). It comprises an extensive dataset with 12,751 unique question-SQL pairs, encompassing 95 large databases. SQL queries are categorized into three difficulty levels, i.e., simple, moderate and challenge. Notably, the SQL queries in the BIRD dataset tend to be more intricate than those in the Spider dataset.

Moreover, our codebase universally supports tuning a wide range of popular dataset, such as WikiSQL (Zhong et al., 2017), CoSQL (Yu et al., 2019), Chase (Guo et al., 2021) (see Appendix A.1 for the detailed statistics of each dataset.) and due to the page limit, we continually post updated experimental results on the project site[2].

**Query-response Construction.** We construct query-response pairs from the datasets so that LLMs can be tuned with (Gao et al., 2023; Xue et al., 2023b). Following Gao et al. (2023), we formulate the pairs using the widely-used *Text Representation Prompt* (Nan et al., 2023) (TRP) format for train, development and test split for all the datasets throughout the experiments.

Shown in Listing 1, TRP represents both schema and query in natural language. In addition, it adds instructions at the very beginning of the prompt to guide LLMs. See Listing 2 and Listing 3 in Appendix A.4 for full examples.

```
1
2 I want you to act as a SQL terminal in front of a database and below is an
  description of the database schema. Write a response that appropriately completes
   the request.
3
4 /* Instruction */
5 Database concert_singer contains tables such as stadium, singer, concert,
  singer_in_concert.
```

---

[2]https://github.com/anonymity-360/DB-GPT-Hub/blob/main/docs/eval_llm_result.md

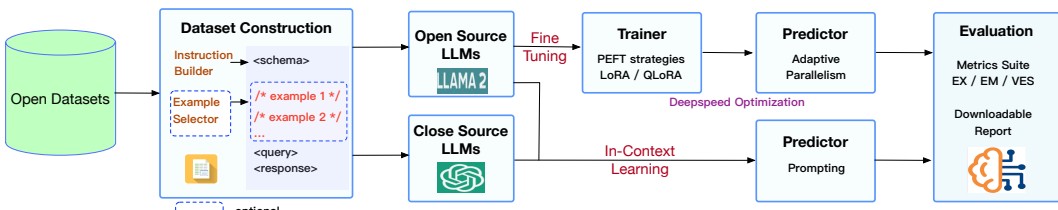

Figure 1: An open benchmarking pipeline using DB-GPT-Hub.

```
 6  Table stadium has columns such as Stadium_ID, Location, Name, Capacity, Highest,
    Lowest, Average. Stadium_ID is the primary key.
 7  Table singer has columns such as Singer_ID, Name, Country, Song_Name,
    Song_release_year, Age, Is_male. Singer_ID is the primary key.
 8  Table concert has columns such as concert_ID, concert_Name, Theme, Stadium_ID,
    Year. concert_ID is the primary key.
 9  Table singer_in_concert has columns such as concert_ID, Singer_ID. concert_ID is
    the primary key.
10  The Stadium_ID of concert is the foreign key of Stadium_ID of stadium.
11  The Singer_ID of singer_in_concert is the foreign key of Singer_ID of singer.
12  The concert_ID of singer_in_concert is the foreign key of concert_ID of concert.
13
14  Please give SQL statement to answer the following question:
15
16  Q: How many singers do we have?
17  Response: SELECT DISTINCT country FROM singer WHERE age  >  20.
18
```

Listing 1: Query-response Pairs in TRP Format on Spider Dataset.

**Metrics.**    We use two commonly used metrics, exact-set-match accuracy (EM), execution accuracy (EX) to evaluate the performance of all models. EM measures the matched SQL keywords between the predicted SQL query and its corresponding ground truth while EX compares the execution output of the predicted SQL query with that of the ground truth SQL query on some database instances. EX provides a more precise estimate of the model's performance since there may be multiple valid SQL queries for a given question. For both metrics, the higher is considered the better. We mainly use EX to evaluate the accuracy of SQLs in the paper. See Appendix A.2 for details.

**Base LLMs.**    We benchmark a range of medium to large-sized LLM variants from 4 prominent LLM families: GLM (Zeng et al., 2022), Qwen (Bai et al., 2023), Baichuan (Baichuan, 2023) and Llama (Touvron et al., 2023).

- ChatGLM3-6B, the up-to-date open version of ChatGLM, an open bilingual language model based on GLM framework.
- Qwen-7B/14B/72B-Chat, a series of aligned models of Qwen.
- Baichuan2-7B/13B-Chat, the up-to-date collection of aligned models of Baichuan.
- LLaMA2-7B/13B/70B-Chat[3], the up-to-date aligned version of LLaMA.
- CodeLLaMA-7B/13B/70B-Instruct, an aligned version of LLaMA-2-13B, tuned with code data.

To ensure a fair comparison, we use the same maximal context length 2048 for all the LLMs. During the evaluation, we leave 512 tokens for response generation. We set the argument temperature as 0 to eliminate the influence of randomness.

**Tuning Methods.**    As the scale of the dataset is notably smaller than that of LLMs, we apply the PEFT strategies –LoRA and QLoRA – to tune the LLMs, respectively. For medium-sized models

---

[3]Due to the page limitation, we have omitted the suffix "-Chat" from the names of LLMs in the tables throughout the following sections. For instance, "Qwen-7B" should be read as "Qwen-7B-Chat" model.

(7B/13B), we adopt 1 Nvidia A100 Tensor Core GPU to run training and inference. For large-sized models (70B), we adopt 8 A100s.

**Benchmark Pipeline.** Figure 1 presents the open benchmarking pipeline implemented in DB-GPT-Hub. This pipeline will facilitate future research in this area and help promote reproducible work.

## 3.2 CODEBASE

To facilitate the innovation of the community, our DB-GPT-Hub contains a well-modularized, easy-to-extend codebase for standardization of implementation, evaluation, and ablation of text-to-SQL methods.

**Software Architecture.** Figure 1 presents the pipeline and architecture of our codebase. Pipelines are decomposed into following parts:

- **Dataset Construction**. Raw text-to-SQL data is processed into a suitable format (e.g., TRF shown in Listing 1 ) to tune LLMs. This includes integrating the schema and database description into a prompt as an instruction, along with various question representations to boost performance during training and evaluation. Additionally, we will select different few-shot strategies, such as example selection and organization, to construct the evaluation dataset Gao et al. (2023).
- **Training.** Our codebase supports the fine-tuning of open-source LLMs with PEFT strategies. We support most of the public architecture with small to large-sized model scales, such as Qwen, Llama, Baichuan, and ChatGLM.
- **Prediction.** Our codebase supports SQL query inference for open-source LLMs with its fine-tuned version and closed-source LLMs as well. We support the few-shot and zero-shot method to generate SQLs for specific scenarios.
- **Evaluation.** Our repository holds different metrics(EX, EM, valid efficiency score(VES)) to evaluate the performance of generated SQL from different perspectives.

**Implementations.** The codebase is built with the PyTorch framework (Paszke et al., 2017), upon the open source project DB-GPT (Xue et al., 2023a; 2024a). We release the code with Apache License 2.0 and we are committed to actively maintain the repository.

## 4 EXPERIMENTS

In this section, with the utility of DB-GPT-Hub, we formally evaluate the text-to-SQL process to determine the performance differences among various LLMs and explore the effect of training paradigms that influence tuning performance of LLMs.

## 4.1 MAIN RESULTS

Table 1 and Table 2 show the evaluation results, measured by EX, on Spider and BIRD datasets, respectively [4]. The results in EM on both datasets can be found in Table 6 and Table 7 in Appendix B.

**Best Models.** Unsurprisingly, tuned CodeLlama families, whose base models haven been optimized for code generation and infilling, show consistently better performance over other competitors on both datasets. Specifically, we have achieved the following key insights:

- As shown in the right-most columns in Table 1 and Table 2, The fine-tuned, small-sized CodeLlama (e.g., CodeLlama-7B-LoRA[5]) exhibits comparable, and in some cases even superior, performance to other tuned medium to large-sized open LLMs, such as Qwen-14B/72B-LoRA.
- CodeLlama-70B-LoRA is universally optimal.

---

[4] For large-sized (70B) models, we found that DeepSpeed optimization is incompatible with QLoRA, so we have left this data blank for the time being.

[5] We use the suffix '-LoRA/QLoRA' to denote the LoRA/QLoRA PEFT strategies applied to tune LLMs, i.e., '-LoRA' means the LLM is tuned with LoRA.

| MODEL | EASY | | MEDIUM | | HARD | | EXTRA | | OVERALL | |
|---|---|---|---|---|---|---|---|---|---|---|
| | BASE | L/QL | BASE | L/QL | BASE | L/QL | BASE | L/QL | BASE | L/QL |
| LLAMA2-7B | 0.000 | 0.887/0.847 | 0.000 | 0.641/0.623 | 0.000 | 0.489/0.466 | 0.000 | 0.331/0.361 | 0.000 | 0.626/0.608 |
| LLAMA2-13B | 0.000 | 0.907/0.911 | 0.000 | 0.729/0.700 | 0.000 | 0.552/0.552 | 0.000 | 0.343/0.319 | 0.000 | 0.680/0.664 |
| LLAMA2-70B | 0.411 | 0.915/– | 0.229 | 0.732/– | 0.190 | 0.560/– | 0.072 | 0.392/– | 0.241 | 0.687/– |
| CODELLAMA-7B | 0.214 | 0.923/0.911 | 0.177 | 0.756/**0.751** | 0.092 | 0.586/0.598 | 0.036 | 0.349/0.331 | 0.149 | 0.702/0.696 |
| CODELLAMA-13B | 0.698 | 0.940/**0.940** | 0.600 | 0.789/0.744 | 0.408 | 0.684/**0.626** | 0.271 | 0.404/0.392 | 0.529 | 0.746/**0.727** |
| CODELLAMA-70B | 0.722 | **0.962**/– | 0.625 | **0.812**/– | 0.443 | **0.716**/– | **0.302** | **0.432**/– | 0.567 | **0.771**/– |
| BAICHUAN2-7B | 0.577 | 0.871/0.891 | 0.352 | 0.630/0.637 | 0.201 | 0.448/0.489 | 0.066 | 0.295/0.331 | 0.335 | 0.603/0.624 |
| BAICHUAN2-13B | 0.581 | 0.903/0.895 | 0.413 | 0.702/0.675 | 0.264 | 0.569/0.580 | 0.187 | 0.392/0.343 | 0.392 | 0.678/0.659 |
| QWEN-7B | 0.395 | 0.855/0.911 | 0.256 | 0.688/0.675 | 0.138 | 0.575/0.575 | 0.042 | 0.331/0.343 | 0.235 | 0.652/0.662 |
| QWEN-14B | **0.871** | 0.895/0.919 | 0.632 | 0.702/0.744 | 0.368 | 0.552/0.598 | 0.181 | 0.367/0.458 | 0.573 | 0.663/0.701 |
| QWEN-72B | 0.831 | 0.927/– | **0.635** | 0.756/– | **0.489** | 0.621/– | 0.277 | 0.367/– | **0.600** | 0.712/– |
| CHATGLM3-6B | 0.000 | 0.855/0.843 | 0.000 | 0.605/0.603 | 0.000 | 0.477/0.506 | 0.000 | 0.271/0.211 | 0.000 | 0.590/0.581 |

Table 1: Evaluations on Spider: EX of base models vs fine-tuned models on each split of complexity and overall dataset. "L" and "QL" denote "LORA" and "QLoRA" tuing methods, respectively.

| MODEL | SIMPLE | | MODERATE | | CHALLENGE | | OVERALL | |
|---|---|---|---|---|---|---|---|---|
| | BASE | L/QL | BASE | L/QL | BASE | L/QL | BASE | L/QL |
| LLAMA2-7B | 0.000 | 0.214/0.211 | 0.000 | 0.108/0.112 | 0.000 | 0.076/0.069 | 0.000 | 0.169/0.168 |
| LLAMA2-13B | 0.000 | 0.226/0.217 | 0.000 | 0.073/0.086 | 0.000 | 0.097/0.069 | 0.000 | 0.167/0.163 |
| LLAMA2-70B | 0.082 | 0.210/– | 0.013 | 0.138/– | 0.014 | 0.126/– | 0.055 | 0.241/– |
| CODELLAMA-7B | 0.010 | 0.299/0.076 | 0.065 | 0.149/0.146 | 0.000 | 0.112/0.128 | 0.085 | 0.237/0.223 |
| CODELLAMA-13B | 0.120 | 0.375/**0.373** | 0.042 | 0.176/**0.179** | 0.042 | 0.141/**0.140** | 0.089 | 0.294/**0.293** |
| CODELLAMA-70B | 0.191 | **0.423**/– | 0.091 | **0.191**/– | **0.063** | **0.159**/– | 0.149 | **0.328**/– |
| BAICHUAN2-7B | 0.051 | 0.231/0.208 | 0.024 | 0.082/0.084 | 0.000 | 0.069/0.105 | 0.038 | 0.171/0.161 |
| BAICHUAN2-13B | 0.048 | 0.0230/0.182 | 0.013 | 0.088/0.067 | 0.021 | 0.111/0.069 | 0.035 | 0.176/0.136 |
| QWEN-7B | 0.035 | 0.235/0.225 | 0.012 | 0.073/0.095 | 0.014 | 0.083/0.082 | 0.023 | 0.171/0.172 |
| QWEN-14B | 0.188 | 0.288/0.252 | 0.049 | 0.136/0.120 | 0.028 | 0.111/0.110 | 0.131 | 0.226/0.198 |
| QWEN-72B | **0.253** | 0.289/– | **0.112** | 0.093/– | 0.048 | 0.083/– | **0.190** | 0.209/– |
| CHATGLM3-6B | 0.000 | 0.204/0.185 | 0.000 | 0.089/0.074 | 0.000 | 0.056/0.042 | 0.000 | 0.156/0.129 |

Table 2: Evaluations on BIRD: EX of base models vs fine-tuned models on each split of complexity and overall dataset. "L" and "QL" denote "LORA" and "QLoRA" tuing methods, respectively.

**Performance Improvement on Tuning.** Table 1 and Table 2 (also shown in Table 9 in Appendix B) illustrate the improvement of PEFT strategies of LLMs on both datasets, highlighting the LLMs proficiency to adapt to high-quality text-to-SQL training data. Notably, tuning yields a larger improvement on Spider compared to BIRD, measured by EX. This suggests that the benefits of tuning become increasingly important in less complex tasks.

| MODEL | EX | | EM | | TIME COST (HOUR) | | GPU MEMORY (GB) | |
|---|---|---|---|---|---|---|---|---|
| | LORA | QLORA | LORA | QLORA | LORA | QLORA | LORA | QLORA |
| LLAMA2-7B | 0.626 | 0.608 | 0.581 | 0.564 | 4.12 | 5.74 | 23.5 | 16.9 |
| LLAMA2-13B | 0.680 | 0.664 | 0.640 | 0.632 | 7.26 | 8.82 | 34.8 | 29.6 |
| CODELLAMA-7B | 0.702 | 0.696 | 0.668 | 0.665 | 4.33 | 6.74 | 23.8 | 16.7 |
| CODELLAMA-13B | 0.746 | 0.727 | 0.701 | 0.682 | 7.26 | 8.82 | 34.8 | 29.6 |
| BAICHUAN2-7B | 0.603 | 0.624 | 0.588 | 0.602 | 3.33 | 7.52 | 20.9 | 11.5 |
| BAICHUAN2-13B | 0.678 | 0.659 | 0.607 | 0.606 | 8.12 | 15.3 | 34.4 | 17.5 |
| QWEN-7B | 0.652 | 0.662 | 0.610 | 0.621 | 2.57 | 6.45 | 28.9 | 17.1 |
| QWEN-14B | 0.663 | 0.701 | 0.658 | 0.665 | 4.23 | 11.32 | 38.4 | 18.1 |

Table 3: The comparison between LoRA and QLoRA on Spider across different perspectives: EX and EM are the performance metrics; the training time and max GPU memory cost are the resource metrics.

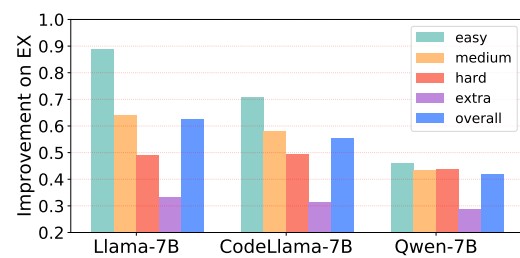

Figure 2: The improvement on tuning with LoRA strategy across subgroups of different complexities.

**Performance for Different SQL Difficulty Levels.**    In Figure 2, using three 7B models for instance, we present the efficacy of tuned LLMs against a spectrum of SQL generation difficulty levels. For all three tuned models, the results highlight that the size of improvement is negatively correlated with the complexities and tuning brings more significant improvement on easy tasks, which reveals the importance of tuning over simpler tasks than difficult ones.

**LoRA vs QLoRA**    We summarize the EX, EM, Time Cost, and GPU memory metrics in Table 3. Firstly, not surprisingly, we see limited differences in generation performance, measured by EX and EM, between models tuned with LoRA and QLoRA. Secondly, consistent with the quantization mechanism, QLoRA takes more time to converge with less GPU memory. For example, compared to Qwen-14B-LoRA, its QLoRA counterpart takes $2\times$ of time with only  $50\%$ GPU memory.

To conclude, in circumstances with restricted computational resources, QLoRA is an efficient tuning alternative that can save memory without sacrificing performance.

### 4.2    ANALYSIS I: FINE-TUNING VS FEW-SHOT PROMPTING

In this subsection, we explore the improvements with tuning compared to few-shot prompting.

**Setup.**    We take two model families –Llama2 and Qwen– and conduct our investigations primarily on the Spider dataset. We use the method *DAIL Selection* (Gao et al., 2023), which currently ranks as the second-best open-source model on the Spider leaderboard, to construct the few-shot prompt. It selects exemplars those have good similarity with both queries and responses, better preserving the mapping in between the query-response pairs.

**Core Insights.**    Due to the page limitation, we put the full results in Table 8 in Appendix B. In both zero-shot and few-shot (1/3/5-shot) evaluation scenarios, tuned LLMs demonstrate superior results, highlighting the LLMs proficiency to adapt to high-quality text-to-SQL training data.

**Effect of Number of Exemplars in Prompting.**    In addition to the superior performances of tuned LLMs, Figure 4 reveals that, for strong (large-sized) models, the EX margin of tuned against base model becomes less prominent on few-shot scenarios. For example, the EX of Qwen-72B-LoRA vs Qwen-72B on 3-shot: 68.5 vs 64.8 and on 5-shot: 68.4 vs 65. This is more clearly observed from a different perspective in Figure 3, where the curves for Qwen-13B/72B is flat at low levels.

This fact is possibly because these Qwen-72B already has strong SQL reasoning capabilities, which has barely been discussed in other text-to-SQL benchmarking works.

In all, fine-tuned models exhibit superior SQL reasoning abilities compared to non-tuned models in few-shot generation scenarios; however, the margin of improvement is relatively small for robust models like Qwen-72B.

**Effect of Model Size.**    From Figure 4, we interpret the few-shot performance w.r.t. the model size for four models (two base models and two tuned models) and observe that:

- Larger models consistently achieve better results in few-shot scenarios compared to their smaller-sized counterparts.

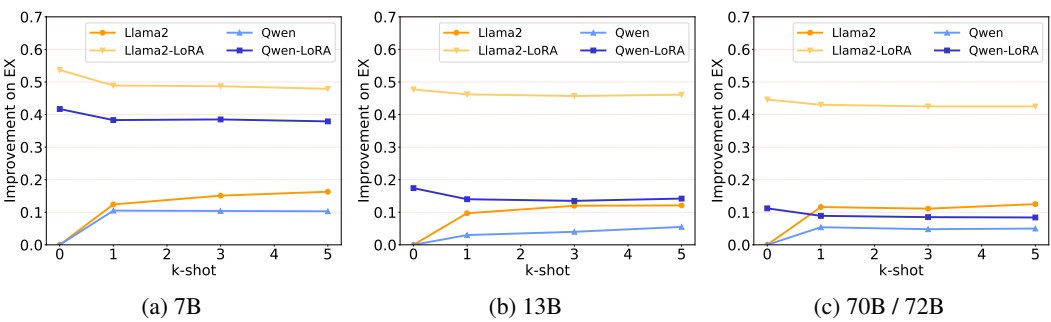

Figure 3: Few-shot evaluations on Spider: EX improvement on few-shot scenarios over zero-shot. EX(k-shot) represents the EX of the target (untuned/tuned) model under k-shot scenario minus EX of the base model in zero-shot scenario, i.e., in (a), Improvement on EX(Qwen-LoRA, 3-shot) = EX(Qwen-LoRA, 3-shot) - EX(Qwen, 0-shot).

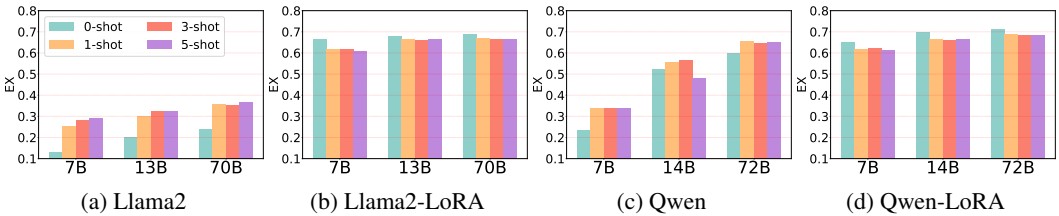

Figure 4: Few-shot evaluations on Spider: the EX performance of Llama2 / Qwen and their tuned counterparts with varying model size.

- For a given few-shot scenario, the performance margin of tuning method over prompting method comes closer when the size of LLMs grows. For example, for 1-shot scenario, the performance improvement on EX of Qwen-LoRA over Qwen is 31.0, 24.1 and 3.5 for 7B, 14B and 72B, respectively.

Recall that the exact figure of few-shot evaluations can be found at Table 8 in Appendix B. Overall, tuning methods continue to outperform prompting methods while the performance gap narrows as the size of the LLMs increases.

### 4.3 ANALYSIS II: FINE-TUNING WITH MORE EXEMPLARS

In this subsection, we explore the possibility of enhancing the performance of LLMs by adding more contextual examples during fine-tuning.

**Setup.** We use Qwen-7B as the base model and construct additional three few-shot (1/3/5-shot) training sets to fine-tune the model. Specifically, the 1/3/5-shot training sets consist of query-response pairs with an additional 1/3/5 exemplars. For a given model, we also evaluate its few-shot performances, same as in section 4.2.

**Core Insights.** Shown in Table 4, we primarily conclude with two insights:

- In a zero-shot evaluation scenario, tuning with additional exemplars does not yield a significant improvement in performance. See the "0-shot" column. This is possible because the training corpus (more examples) mismatches the evaluation setting (no examples).

- In 1/3/5-shot evaluation scenarios, adding more contextual examples contributes to the notable improvement over the counterpart tuned with 0-shot training corpus. It means that the performance loss on few-shot evaluation for zero-shot training is caused by the prompt mismatch of training and evaluation dataset.

| MODEL | 0-SHOT | | 1-SHOT | | 3-SHOT | | 5-SHOT | |
|---|---|---|---|---|---|---|---|---|
| | EM | EX | EM | EX | EM | EX | EM | EX |
| QWEN-7B | 16.1 | 22.9 | 27.4 | 34.0 | 27.6 | 33.9 | 25.9 | 33.8 |
| QWEN-7B-LoRA (0-SHOT) | 61.0 | **65.3** | 58.4 | 61.8 | 57.8 | 62.0 | 57.7 | 61.4 |
| QWEN-7B-LoRA (1-SHOT) | 61.2 | 64.0 | 61.7 | **64.8** | 60.8 | **63.8** | 61.8 | **64.8** |
| QWEN-7B-LoRA (3-SHOT) | 61.0 | 62.8 | 62.0 | 62.8 | 60.7 | 62.1 | 60.7 | 62.9 |
| QWEN-7B-LoRA (5-SHOT) | 60.4 | 62.7 | 62.0 | 64.0 | 61.5 | 63.2 | 60.9 | 63.5 |
| QWEN-7B-LoRA (RANDOM-SHOT) | **61.5** | 63.0 | **62.1** | 64.0 | **62.2** | 63.6 | **61.9** | 63.6 |

Table 4: Few-shot Evaluations on Spider: EM and EX of fine-tuned models with the different number of examples in the training corpus.

• The *random-shot* strategy, which refers to randomly adding 0/1/3/5 examples into the training corpus, achieves the highest EM scores. This finding is consistent with that proposed by (Sun et al., 2023): diverse training corpus benefits the fine-tuning of LLMs.

## 5 RELATED WORK

### 5.1 LLM-EMPOWERED TEXT-TO-SQL METHODS

Driven by the considerable success of LLMs, the field of LLM-empowered text-to-SQL has captured the interest of a large amount of researchers both in nature language process and database community recently. The models on LLM-based text-to-SQL can be categorized into supervised fine-tuning based and prompting based methods. Popular fine-tuned text-to-sql models are SQL-PaLM (Sun et al., 2023), PICARD (Scholak et al., 2021) and RESDSQL (Li et al., 2023a). In contrast to supervised fine-tuned models, prompting-based models do not require additional fine-tuning on task-specific training data. Instead, they solely rely on the zero-shot and few-shot (Rajkumar et al., 2022; Liu et al., 2023) capabilities inherent in LLMs. Within the prompting paradigm, the pivotal factor for query representation lies in the design of the prompt (Wei et al., 2022; Zhou et al., 2022; Wang et al., 2022a). In particular, DIN-SQL (Pourreza & Rafiei, 2023) introduces adaptive prompt strategies via task decomposition to effectively address challenges associated with schema linking. DAIL-SQL (Gao et al., 2023) proposes a refined prompt selection and organization strategy to improve the performance. In DB-GPT-Hub, we offer scripts to support researchers in fine-tuning LLMs in accordance with the methodologies established in SQL-PaLM. In addition, we also integrate the popular prompt techniques used in DAIL-SQL.

### 5.2 TEXT-TO-SQL BENCHMARKS

A pivotal factor in the progression of text-to-SQL is the establishment of high-quality benchmarks. Early benchmarks focus on single databases, including ATIS (Dahl et al., 1994), GeoQuery (Zelle & Mooney, 1996), Academic (Li & Jagadish, 2014), Advising (Finegan-Dollak et al., 2018), and more recent additions such as SEDE (Hazoom et al., 2021) and MIMICSQL (Wang et al., 2019). These benchmarks and datasets are often adapted from real-life applications, with many containing domain-specific knowledge that may not generalize effectively to unseen SQL domains. Hence, large-scale cross-domain datasets featuring professional SQL queries, such as Squall (Shi et al., 2020), Spider (Yu et al., 2018a), Spider-Syn (Gan et al., 2021), WikiSQL (Zhong et al., 2017), and SparC (Yu et al., 2020), have been introduced to facilitate comprehensive method analyses.

In retrospect, we realize two concurrent works (Gao et al., 2023; Zhang et al., 2024) which perform systematical benchmarking on text-to-SQL methods. Important distinctions of their work from ours include: 1. *comprehensiveness of benchmark settings*: we evaluate both ICL and **medium to large-sized** fine-tuning methods in an end-to-end manner while Gao et al. (2023) focus on ICL methods and Zhang et al. (2024) assess various sub-tasks of the text-to-SQL process; 2. *open source of the codebase*: we released a well-maintained open repository on Github containing all code and data assets, which, to the best of knowledge, is one of the most popular text-to-SQL benchmark repositories (over 1k stars so far), while neither of them has achieved this.

## 6 CONCLUSION

In this study, we conduct a systematic benchmarking of the various LLMs within the text-to-SQL pipeline. Our benchmarking provides a meticulous perspective on the pipeline, equipping the research community with strategies to improve the semantic understanding of LLMs.

## 7 LIMITATIONS

The large computational resources required for LLM training might not be accessible to all researchers and practitioners, which may limit the reproducibility of our findings.

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

# Appendices

## A    EXPERIMENTAL DETAILS

### A.1    DATASET DETAILS

**Spider** (Yu et al., 2018b). It consists of 10,181 questions and 5,693 unique complex SQL queries across 200 databases, covering 138 domains, each containing multiple tables. The standard protocol for this dataset divides it into 8,659 training examples across 146 databases, 1,034 development examples across 20 databases, and a holdout of 2,147 test examples across 34 databases. The databases used in each of these sets are nonoverlapping. SQL queries are categorized into four difficulty levels, based on the number of SQL keywords used, the presence of nested subqueries, and the usage of column selections and aggregations.

**BIRD** (Li et al., 2023b). This dataset represents a pioneering, cross-domain dataset that examines the impact of extensive database contents on text-to-SQL parsing. BIRD contains over 12,751 unique question-SQL pairs, 95 big databases with a total size of 33.4 GB. It also covers more than 37 professional domains, such as blockchain, hockey, healthcare and education, etc. BIRD also introduces external knowledge as an additional resource to assist models in generating accurate SQL queries. Specifically four sources of external knowledge were introduced: numeric reasoning knowledge, domain knowledge, synonym knowledge, and value illustration. Notably, the SQL queries in the BIRD dataset tend to be more intricate than those in the Spider dataset.

**WikiSQL** (Zhong et al., 2017). This dataset consists of a corpus of 80,654 natural statement expressions and sql annotations of 24,241 tables. Each query in WikiSQL is limited to the same table and does not contain complex operations such as sorting, grouping. The queries in WikiSQL are limited to the same table and do not include complex operations such as sorting, grouping, subqueries, etc.

**CoSQL** (Yu et al., 2019). This dataset is a conversational version of the Spider task. CoSQL consists of 30,000 rounds and 10,000 annotated SQL queries from Wizard-of-Oz's collection of 3k conversations querying 200 complex databases across 138 domains. Each conversation simulates a realistic DB query scenario in which a staff member explores the database as a user and a SQL expert uses SQL to retrieve answers, clarify ambiguous questions, or otherwise inform.

**Chase** (Guo et al., 2021). This data is to date the largest Chinese dataset for the cross-database context-dependent Text-to-SQL problem. It consists of 5,459 question sequences (17,940 questions) over 280 databases. Each question in Chase has rich semantic annotations, including its SQL query, contextual dependency, and schema linking.

### A.2    METRICS DETAILS

We clarify the properties of the two metrics in details.

**Exact-set match accuracy** (EM). EM treats each clause as a set and compares the prediction for each clause to its corresponding clause in the reference query. A predicted SQL query is considered correct only if all of its components match the ground truth. EM does not take values into account.

**Execution accuracy** (EX). EX compares the execution output of the predicted SQL query with that of the ground truth SQL query on some database instances. Execution accuracy provides a more precise estimate of the performance of the method as there may be multiple valid SQL queries for a given question while EM only evaluates the predicted SQL against one of them.

### A.3    IMPLEMENTATION DETAILS

All models are implemented using the PyTorch framework (Paszke et al., 2017). For parameter scale with 7B and 13B models, we adopt 1 Nvidia A100 Tensor Core GPU to run training. For the parameter scale of 70B model, we adopt 8*A100 to run training and inference.

**Fine-tuning hyperparameters setting**    The hyperparameters of the training are shown in Table 5.

| Parameter | 7B | 13B | 70B |
|---|---|---|---|
| GPUs | 1*A100 | 1*A100 | 8*A100 |
| max source length | 2048 | 2048 | 2048 |
| max target length | 512 | 512 | 512 |
| fine-tuning type | lora | lora | lora |
| lora rank | 64 | 64 | 64 |
| lora alpha | 32 | 32 | 32 |
| lr | 0.0002 | 0.0002 | 0.0002 |
| epoch | 8 | 8 | 8 |

Table 5: Parameter setting of fine tuning for different model scale

## A.4 FEW SHOT PROMPTING

```
20 Given the following database schema :
21
22
23 Table advisor, columns = [*,s_ID,i_ID]
24 Table classroom, columns = [*,building,room_number,capacity]
25 Table course, columns = [*,course_id,title,dept_name,credits]
26 Table department, columns = [*,dept_name,building,budget] Table instructor,
   columns = [*,ID,name,dept_name,salary] Table prereq, columns = [*,course_id,
   prereq_id]
27 Table section, columns = [*,course_id,sec_id,semester,year,building,room_number,
   time_slot_id]
28 Table student, columns = [*,ID,name,dept_name,tot_cred] Table takes, columns = [*,
   ID,course_id,sec_id,semester,year,grade]
29 Table teaches, columns = [*,ID,course_id,sec_id,semester,year]
30 Table time_slot, columns = [*,time_slot_id,day,start_hr,start_min,end_hr,end_min]
31
32 Please write queries to answer the following questions:
33
34 Q: Find the title of courses that have two prerequisites.
35 Response: SELECT T1.title FROM course AS T1 JOIN prereq AS T2 ON T1.course_id =
   T2.course_id GROUP BY T2.course_id HAVING count(*) = 2.
36
37 Q: Find the room number of the rooms which can sit 50 to 100 students and their
   buildings.
38 Response: SELECT building , room_number FROM classroom WHERE capacity BETWEEN 50
   AND 100.
39
40 Q: Give the name of the student in the History department with the most credits.
41 Response: SELECT name FROM student WHERE dept_name = 'History' ORDER BY tot_cred
   DESC LIMIT 1.
42
43 Q: Find the total budgets of the Marketing or Finance department.
44 Response: SELECT sum(budget) FROM department WHERE dept_name = 'Marketing' OR
   dept_name = 16 'Finance'.
45
46 Q: Find the department name of the instructor whose name contains 'Soisalon'.
47 Response: SELECT dept_name FROM instructor WHERE name LIKE '%Soisalon%'.
48
49 Q: What is the name of the department with the most credits?
50 Response: SELECT dept_name FROM course GROUP BY dept_name ORDER BY sum(credits)
   DESC LIMIT 1.
51
52 Q: How many instructors teach a course in the Spring of 2010?
53 Response: SELECT COUNT (DISTINCT ID) FROM teaches WHERE semester = 'Spring' AND
   YEAR = 2010.
54
55 Q: Find the name of the students and their department names sorted by their total
    credits in ascending order.
56 Response: SELECT name, dept_name FROM student ORDER BY tot_cred.
```

```
57
58 Q: Find the year which offers the largest number of courses.
59 Response: SELECT YEAR FROM SECTION GROUP BY YEAR ORDER BY count(*) DESC LIMIT 1.
60
61 Q: What are the names and average salaries for departments with average salary
   higher than 42000?
62 Response: SELECT dept_name , AVG (salary) FROM instructor GROUP BY dept_name
   HAVING AVG (salary) > 42000.
63
64 Q: Find the minimum salary for the departments whose average salary is above the
   average payment of all instructors.
65 Response: SELECT min(salary) , dept_name FROM instructor GROUP BY dept_name HAVING
    avg(salary) > (SELECT avg(salary) FROM instructor).
66
67 Q: What is the course title of the prerequisite of course Mobile Computing?
68 Response: SELECT title FROM course WHERE course_id IN (SELECT T1.prereq_id FROM
   prereq AS T1 JOIN course AS T2 ON T1.course_id = T2.course_id WHERE T2.title = '
   Mobile Computing').
69
70
71 Q: Give the title and credits for the course that is taught in the classroom with
    the greatest capacity.
72 Response: SELECT T3.title , T3.credits FROM classroom AS T1 JOIN SECTION AS T2 ON
    T1.building = T2.building AND T1.room_number = T2.room_number JOIN course AS T3
   ON T2.course_id = T3.course_id WHERE T1.capacity = (SELECT max(capacity) FROM
   classroom).
73
74 Q: Find the name of students who took any class in the years of 2009 and 2010.
75 Response: SELECT DISTINCT T1.name FROM student AS T1 JOIN takes AS T2 ON T1.id =
   T2.id WHERE T2.YEAR = 2009 OR T2.YEAR = 2010.
76
77 Q: Find the total number of students and total number of instructors for each
   department.
78 Response: SELECT count(DISTINCT T2.id) , count(DISTINCT T3.id) , T3.dept_name
   FROM department AS T1 JOIN student AS T2 ON T1.dept_name = T2.dept_name JOIN
   instructor AS T3 ON T1.dept_name = T3.dept_name GROUP BY T3.dept_name.
79
80 Q: Find the buildings which have rooms with capacity more than 50.
81 Response: SELECT DISTINCT building FROM classroom WHERE capacity > 50
```

Listing 2: Full Examples of Text Representation Prompt on Spider Dataset.

```
82 Given the following database schema :
83
84 Table movies, columns = [*,movie_id,movie_title,movie_release_year,movie_url,
   movie_title_language,movie_popularity,movie_image_url,director_id,director_name,
   director_url]
85 Table ratings, columns = [*,movie_id,rating_id,rating_url,rating_score,
   rating_timestamp_utc,critic,critic_likes,critic_comments,user_id,user_trialist]
86 Table lists, columns = [*,user_id,list_id,list_title,list_movie_number,
   list_update_timestamp_utc,list_creation_timestamp_utc,list_followers,list_url,
   list_comments,list_description]
87 Table lists_users, columns = [*,user_id,list_id,list_update_date_utc,
   list_creation_date_utc,user_trialist,user_subscriber,user_avatar_image_url,
   user_cover_image_url,user_eligible_for_trial,user_has_payment_method]
88
89 Please write queries to answer the following questions:
90 Q: How many movies in the database were directed by Christopher Nolan?
91 Response: SELECT COUNT(*) FROM movies WHERE director_name = 'Christopher Nolan'.
92
93 Q: List all movies that have a popularity greater than 5000.
94 Response: SELECT movie_title FROM movies WHERE movie_popularity > 5000.
95
96 Q: Retrieve the URL of the most popular movie.
```

```
97   Response: SELECT movie_url FROM movies ORDER BY movie_popularity DESC LIMIT 1;
98
99   Q: Which user IDs have rated a movie on the 1st of January 2021.
100  Response: SELECT DISTINCT user_id FROM ratings WHERE rating_timestamp_utc BETWEEN
     '2021-01-01 00:00:00' AND '2021-01-01 23:59:59';.
101
102  Q: What are the names of the directors for movies that have an image URL
     containing 'poster'?
103  Response: SELECT DISTINCT director_name FROM movies WHERE movie_image_url LIKE '%
     poster%'.
104
105  Q: Give me the IDs and release years of movies that have both a rating score
     higher than 4 and have been included in at least 10 lists created by users who
     had a payment method when they created the list.
106  Response: SELECT m.movie_id, m.movie_release_year FROM movies m JOIN ratings r ON
     m.movie_id = r.movie_id JOIN lists_users lu ON lu.user_id = ANY(SELECT user_id
     FROM lists WHERE list_id IN (SELECT list_id FROM lists WHERE movie_id = m.movie_id
     )) WHERE r.rating_score > 4 AND lu.user_has_payment_method = 1 GROUP BY m.movie_id
     , m.movie_release_year HAVING COUNT(DISTINCT lu.list_id) >= 10.
107
108  Q: Find the title of the most popular movie among those that have never received
     any critic comments.
109  Response: SELECT movie_title FROM movies JOIN ratings ON movies.movie_id = ratings
     .movie_id WHERE critic_comments = 0 ORDER BY movie_popularity DESC LIMIT 1;
110
111  Q: Find the names of movies from the year 2000 which have been added to at least
     5 different lists and have an image URL available.
112  Response: SELECT DISTINCT m.movie_title FROM movies m JOIN lists l ON m.movie_id
     IN (SELECT movie_id FROM lists WHERE list_id = l.list_id) WHERE m.
     movie_release_year = 2000 AND m.movie_image_url IS NOT NULL GROUP BY m.movie_id
     HAVING COUNT(DISTINCT l.list_id) >= 5.
113
114  Q: Which user created the most number of lists while being a subscriber and
     having a profile cover image?
115  Response: SELECT user_id, COUNT(list_id) as num_lists FROM lists_users WHERE
     user_subscriber = 1 AND user_cover_image_url IS NOT NULL GROUP BY user_id ORDER BY
      num_lists DESC LIMIT 1.
116
117  Q: Provide the critic made by users who rated a movie more than 3 but less than 5
      and got at least 10 likes on their review.
118  Response: SELECT critic FROM ratings WHERE rating_score BETWEEN 3 AND 5 AND
     critic_likes >= 10.
119
120  Q: How many lists were created by users who were subscribers and not trialists on
      January 1st, 2020?
121  Response: SELECT COUNT(DISTINCT list_id) FROM lists_users WHERE user_subscriber =
     1 AND user_trialist = 0 AND list_creation_date_utc = '2020-01-01'.
122
123  Q: What are the titles of the lists which were created on '2022-05-15' and have
     more than 50 comments?
124  Response: SELECT list_title FROM lists WHERE list_creation_timestamp_utc =
     '2022-05-15' AND list_comments > 50.
125
126
127  Q: What is the name and URL of the movie that has the latest rating timestamp?
128  Response: SELECT movie_title, movie_url FROM movies WHERE movie_id = (SELECT
     movie_id FROM ratings ORDER BY rating_timestamp_utc DESC LIMIT 1).
129
130  Q: Which movie has the highest number of critic likes.
131  Response: SELECT movie_id FROM ratings ORDER BY critic_likes DESC LIMIT 1;
132
133  Q: Retrieve the list description and URL for lists created by trialists that have
      been updated since 2021 and contain movies directed by Christopher Nolan.
134  Response: SELECT l.list_description, l.list_url FROM lists l JOIN lists_users lu
     ON l.list_id = lu.list_id JOIN movies m ON m.movie_id IN (SELECT movie_id FROM
```

| MODEL | EASY | | MEDIUM | | HARD | | EXTRA | | OVERALL | |
|---|---|---|---|---|---|---|---|---|---|---|
| | BASE | L/QL | BASE | L/QL | BASE | L/QL | BASE | L/QL | BASE | L/QL |
| LLAMA2-7B | 0.000 | 0.827/0.810 | 0.000 | 0.614/0.574 | 0.000 | 0.408/0.443 | 0.000 | 0.307/0.295 | 0.000 | 0.581/0.564 |
| LLAMA2-13B | 0.000 | 0.867/0.835 | 0.000 | 0.670/0.670 | 0.000 | 0.483/0.517 | 0.000 | 0.386/0.349 | 0.000 | 0.640/0.632 |
| LLAMA2-70B | 0.327 | 0.847/— | 0.112 | 0.679/— | 0.075 | 0.454/— | 0.018 | 0.382/— | 0.142 | 0.635/— |
| CODELLAMA-7B | 0.174 | 0.883/0.871 | 0.127 | 0.736/**0.721** | 0.063 | 0.523/0.553 | 0.012 | 0.309/0.291 | 0.121 | 0.643/0.628 |
| CODELLAMA-13B | 0.617 | 0.910/**0.910** | 0.545 | 0.727/0.688 | 0.377 | 0.624/**0.556** | 0.224 | 0.365/0.382 | 0.487 | 0.706/**0.682** |
| CODELLAMA-70B | 0.688 | **0.928**/— | **0.582** | **0.723**/— | **0.400** | **0.655**/— | **0.278** | **0.366**/— | **0.527** | **0.713**/— |
| BAICHUAN2-7B | 0.326 | 0.832/0.815 | 0.104 | 0.588/0.621 | 0.025 | 0.402/0.454 | 0.000 | 0.225/0.286 | 0.119 | 0.579/0.602 |
| BAICHUAN2-13B | 0.363 | 0.839/0.827 | 0.141 | 0.632/0.650 | 0.040 | 0.483/0.460 | 0.000 | 0.325/0.313 | 0.155 | 0.607/0.606 |
| QWEN-7B | 0.365 | 0.802/0.778 | 0.101 | 0.643/0.608 | 0.063 | 0.517/0.471 | 0.024 | 0.331/0.313 | 0.161 | 0.610/0.578 |
| QWEN-14B | **0.758** | 0.867/0.851 | 0.318 | 0.713/0.735 | 0.172 | 0.529/0.506 | 0.066 | 0.398/0.367 | 0.359 | 0.623/0.668 |
| QWEN-72B | 0.754 | 0.903/— | 0.316 | 0.726/— | 0.241 | 0.523/— | 0.102 | 0.386/— | 0.374 | 0.680/— |
| CHATGLM3-6B | 0.000 | 0.776/0.763 | 0.000 | 0.564/0.533 | 0.000 | 0.457/0.477 | 0.000 | 0.261/0.224 | 0.000 | 0.521/0.542 |

Table 6: Evaluations on Spider: EM of base models vs fine-tuned models on each split of complexity and overall dataset. "L" and "QL" denote "LORA" and "QLoRA" tuing methods, respectively.

| MODEL | SIMPLE | | MODERATE | | CHALLENGE | | OVERALL | |
|---|---|---|---|---|---|---|---|---|
| | BASE | L/QL | BASE | L/QL | BASE | L/QL | BASE | L/QL |
| LLAMA2-7B | 0.000 | 0.068/0.062 | 0.000 | 0.015/0.017 | 0.000 | 0.000/0.000 | 0.000 | 0.046/0.043 |
| LLAMA2-13B | 0.000 | 0.115/0.087 | 0.000 | 0.013/0.017 | 0.000 | 0.069/0.000 | 0.000 | 0.074/0.058 |
| LLAMA2-70B | 0.000 | 0.107/— | 0.000 | 0.028/— | 0.000 | 0.000/— | 0.000 | 0.072/— |
| CODELLAMA-7B | 0.000 | 0.228/0.059 | 0.000 | 0.089/0.086 | 0.000 | 0.058/0.062 | 0.000 | 0.128/0.119 |
| CODELLAMA-13B | 0.088 | 0.293/**0.346** | 0.000 | 0.129/**0.136** | 0.000 | 0.112/**0.124** | 0.029 | 0.256/**0.243** |
| CODELLAMA-70B | 0.102 | **0.348**/— | **0.059** | 0.124/— | **0.032** | **0.087**/— | **0.082** | **0.255**/— |
| BAICHUAN2-7B | 0.000 | 0.078/0.068 | 0.000 | 0.022/0.017 | 0.000 | 0.000/0.000 | 0.000 | 0.054/0.046 |
| BAICHUAN2-13B | 0.010 | 0.073/0.056 | 0.000 | 0.004/0.018 | 0.000 | 0.014/0.000 | 0.035 | 0.045/0.037 |
| QWEN-7B | 0.000 | 0.067/0.082 | 0.000 | 0.010/0.015 | 0.000 | 0.007/0.013 | 0.000 | 0.043/0.055 |
| QWEN-14B | 0.000 | 0.089/0.084 | 0.000 | 0.028/0.021 | 0.000 | 0.014/0.021 | 0.000 | 0.064/0.059 |
| QWEN-72B | **0.154** | 0.243/— | 0.023 | 0.048/— | 0.012 | 0.038/— | 0.042 | 0.089/— |
| CHATGLM3-6B | 0.000 | 0.124/0.112 | 0.000 | 0.045/0.048 | 0.000 | 0.026/0.028 | 0.000 | 0.068/0.051 |

Table 7: Evaluations on BIRD: EM of base models vs fine-tuned models on each split of complexity and overall dataset. "L" and "QL" denote "LORA" and "QLoRA" tuing methods, respectively.

```
lists WHERE list_id = l.list_id) WHERE lu.user_trialist = 1 AND l.
list_update_timestamp_utc > '2021-01-01' AND m.director_name = 'Christopher Nolan
'.
135
136 Q: List all the directors along with the average rating score for movies they
directed that have over 1000 followers on Mubi lists.
137 Response: SELECT director_name, AVG(rating_score) AS avg_rating FROM movies JOIN
ratings ON movies.movie_id = ratings.movie_id LEFT JOIN lists ON movies.movie_id =
lists.list_movie_number GROUP BY director_name HAVING SUM(list_followers) > 1000.
```

Listing 3: Full Examples of Text Representation Prompt on BIRD Dataset.

# B  MORE EXPERIMENT RESULT

## B.1  EM METRICS OF SPIDER DATASET

The EM metric of BIRD dataset are show in Table 6.

## B.2  EM METRIC OF BIRD DATASET

The EM metric of BIRD dataset are show in Table 7.

| MODEL | 0-SHOT | | 1-SHOT | | 3-SHOT | | 5-SHOT | |
|---|---|---|---|---|---|---|---|---|
| | EM | EX | EM | EX | EM | EX | EM | EX |
| LLAMA2-7B | 3.1 | 13.0 | 18.5 | 25.4 | 22.1 | 28.1 | 22.6 | 29.3 |
| LLAMA2-7B-LORA | **63.9** | **66.7** | **58.5** | **61.9** | **59.8** | **61.7** | **58.9** | **60.9** |
| LLAMA2-13B | 2.4 | 20.3 | 13.2 | 30.0 | 15.5 | 32.3 | 16.2 | 32.4 |
| LLAMA2-13B-LORA | **62.7** | **67.0** | **62.5** | **66.5** | **60.6** | **66.0** | **61.3** | **66.4** |
| LLAMA2-70B | 14.2 | 24.1 | 24.8 | 35.7 | 25.4 | 35.2 | 27.7 | 36.6 |
| LLAMA2-70B-LORA | **66.3** | **68.7** | **62.8** | **67.1** | **61.6** | **66.6** | **61.5** | **66.6** |
| QWEN-7B | 16.1 | 23.5 | 27.4 | 34.0 | 27.6 | 33.9 | 25.9 | 33.8 |
| QWEN-7B-LORA | **61.0** | **65.2** | **58.4** | **61.8** | **57.8** | **62.0** | **57.5** | **61.4** |
| QWEN-14B | 32.3 | 52.4 | 40.4 | 55.4 | 43.4 | 56.4 | 44.8 | 57.9 |
| QWEN-14B-LORA | **67.8** | **69.8** | **64.5** | **66.4** | **64.3** | **65.9** | **64.3** | **66.6** |
| QWEN-72B | 37.4 | 60.0 | 51.5 | 65.4 | 51.3 | 64.8 | 51.3 | 65.0 |
| QWEN-72B-LORA | **68.0** | **71.2** | **65.1** | **68.9** | **65.5** | **68.5** | **64.2** | **68.4** |

Table 8: Few shot evaluations on Spider: base models vs fine-tune models.

| MODEL | SPIDER | | BIRD | |
|---|---|---|---|---|
| | LoRA | QLoRA | LoRA | QLoRA |
| LLAMA2-7B | ↑0.626 | ↑0.608 | ↑0.169 | ↑0.168 |
| LLAMA2-13B | ↑0.680 | ↑0.664 | ↑0.167 | ↑0.163 |
| LLAMA2-70B | ↑0.687 | - | ↑0.186 | — |
| CODELLAMA-7B | ↑0.453 | ↑0.447 | ↑0.228 | ↑0.214 |
| CODELLAMA-13B | ↑0.217 | ↑0.198 | ↑0.204 | ↑0.204 |
| CODELLAMA-70B | ↑0.204 | — | ↑0.179 | — |
| BAICHUAN2-7B | ↑0.268 | ↑0.289 | ↑0.133 | ↑0.123 |
| BAICHUAN2-13B | ↑0.286 | ↑0.267 | ↑0.141 | ↑0.101 |
| QWEN-7B | ↑0.417 | ↑0.427 | ↑0.148 | ↑0.133 |
| QWEN-14B | ↑0.090 | ↑0.128 | ↑0.075 | ↑0.068 |
| QWEN-72B | ↑0.112 | — | ↑0.019 | — |
| CHATGLM3-6B | ↑0.590 | ↑0.581 | ↑0.156 | ↑0.128 |

Table 9: Evaluations on Spider and BIRD: EX improvement on tuning with LoRA / QLoRA over base model.

### B.3 MORE RESULTS ON FEW-SHOT EVALUATION

The execution accuracy of k-shots prompt on different models with it's fine-tuned version are shown in Table 8

### B.4 LORA AND QLORA

The performance improvement of LoRA and QLoRA on Spider and BIRD are shown in Table 9

## C ONGOING AND FUTURE WORK

We are currently exploring several extensions to deal with more complex dialogue and analytics cases in our system. We are particularly interested in handling

- More powerful agents. Users may want our system not only to perform the analysis but also provide more powerful abilities on text-to-SQL, such as sequential predictions (Jin et al., 2023; Xue et al., 2024b) based on historical data and predictive decision abilities (Pan et al., 2023).

- Integration of more model training techniques. In addition to pre-training, the community is also interested in continual learning techniques for language models, such as continual pre-training (Jiang et al., 2023), prompt learning (Wang et al., 2022b) or positional encoding techniques (Zhu et al., 2024). The integration of these methods will greatly facilitate the research community in these areas.

