# OpenReview forum: "DB-GPT-Hub: Towards Open Benchmarking Text-to-SQL Empowered by Large Language Models"
_ICLR.cc/2025/Conference — ICLR 2025 Conference Withdrawn Submission_

### Official Review · Reviewer_D9WG · 2024-10-19

**Soundness:** 3
**Presentation:** 1
**Contribution:** 1
**Rating:** 3
**Confidence:** 5

**Summary:**

This paper investigates capabilities of partial open-source language models across different size on SPIDER and BIRD text-to-SQL benchmarks by fine-tuning approach. Authors summarize and provide an unified training and evaluation pipelines. The experiment summarizes phenomenon and insights for text-to-SQL in PEFT.

**Strengths:**

1. Authors summarize and curate an unified codebase for LLM training and evaluation on many popular benchmarks such as BIRD and SPIDER.
2. Authors conduct experiments across various model sizes and settings including zero-shot and few-shot settings. Also authors implemented lora and Q-lora of partial open-source models on these benchmarks.

**Weaknesses:**

In general, the motivations, contributions and paper writing are not good enough for readers to follow. It's hard to follow the paper to see why this work matters in the big scope. The new ideas aren't spelled out clearly, and the writing could use some polish.
## Motivations and Introduction:
1) The logic flow of Introduction in lines 35-37 is unclear. The authors cite seq-to-seq methods as important, but then suddenly discuss decoding techniques without a clear connection. In the paper that authors cite, seq-to-seq modeling involve encoder and decoder usually such as [1] [2]. This jump is confusing, especially since the paper focuses on decoder-only models, not seq-to-seq architectures. If the authors aim to show a progression from seq-to-seq to decoder-only models for SQL generation, they should explain this transition and its rationale. This would help readers better understand the context and relevance of the discussed techniques.

2) The second paragraph is unclear as well. The use of **"While"** suggests a contrast in two clauses, but the relationship between LLMs' improved capabilities and their emergence as a new paradigm seems more **causal** than contrastive. A clearer structure might be: "As LLMs have become more powerful, their application to text-to-SQL tasks has shown promise, as evidenced by [work1, work2, ...]." The claim that LLM-based methods are dominating since 2023 needs stronger support. Including more recent citations (e.g., DIN-SQL, DAIL-SQL) or providing a link to the SPIDER / BIRD leaderboard would be more useful for readers to follow. Additionally, authors mentioned human-like communication, but this seems not related to the focus of papers on single-turn text-to-SQL benchmarks like SPIDER and BIRD. The relevance of communication to these tasks should be clarified or this point reconsidered.

3) In lines 48-49, the phrase **"ambiguity and complexity"** lacks specificity in this context. It's unclear whether the authors are referring to ambiguity in the natural language questions, database schemas, schema linking, SQL generation, or some combination of these elements. Without more precise details, readers may struggle to understand the exact challenges being addressed. The authors should clarify which aspects of the text-to-SQL task they find ambiguous or complex, providing specific examples if possible. And in this paper which aspects they will discuss.

4) The argument about human effort lacks consistency. They claim in-context learning requires huge expert annotation, but their study uses only 7 examples for it. In contrast, fine-tuning datasets like SPIDER and BIRD contain > 10k examples. This discrepancy undermines the motivation for focusing on fine-tuning approaches. The entire paragraph also presents a confusing narrative. It begins by criticizing in-context learning, then shifts to praising fine-tuning, suggests fine-tuning is under-explored, and abruptly returns to highlighting the advantages of in-context learning. The logic is inconsistent and difficult to follow. The authors need to clarify their position and present a coherent argument about their research focus and its significance. Without a clear progression of ideas, readers may struggle to understand the paper motivation and objectives.

5) The phrase "universally acknowledged" in Line 57 is vague and unsupported. In academic writing, such claims should be backed by concrete evidence, such as citation counts, systematic reviews, or formal user studies. Similarly, the term "relatively smaller" is imprecise without a clear point of comparison. Please define a range such as < 13B or somehow. Without specific definitions or references, these size comparisons remain ambiguous. The authors should provide clear, quantifiable metrics or cite relevant literature to support their claims about model acknowledgment and size categories.

6) The motivation for this work is unclear. The authors try to convey the message that benchmarks for evaluating fine-tuning capabilities of LLMs on text-to-SQL tasks are missing and important. However, this claim is puzzling given the existence of SPIDER and BIRD datasets, which already include training sets which are used for fine-tuning. This paper does **not** introduce new datasets, metrics, or evaluation methodologies, making it difficult to discern its novel contributions, particularly in terms of benchmarking. Without clear advancements in dataset creation or evaluation techniques, the significance of this work in the context of existing benchmarks is questionable. The authors need to articulate more clearly what gap their research fills and how it advances the field beyond the capabilities of current benchmarks.

7) Also, I think one of the biggest problems of this paper is overstatement. All features listed in bullet points cannot be proven in the rest of the paper. Here are in-depth reasons:
   - Standardization: Among the whole paper including appendices, I cannot quite understand how authors standardize pipeline. The author states that their standardized pipelines are **transparent and consistent**. What does **"transparent and consistent"** mean? And why and how are previous works not "transparent and consistent"? Whether this is a better statement by saying what and why previous works are not "transparent and consistent" first, and how do you fix this issue? Also how many contributions of this standardization of your work are? For example, SPIDER and BIRD share similar settings. All them have json as input, sqlite as database formats. Therefore, whether your efficient standardization stems from efforts of these original benchmarks **already**? Or if it's not, please make your contributions clear by stating the difficulties and how you resolve them in the main content since you consider this as one of main contributions here.
   - Comprehensive: I suggest authors could list their aspects of **comprehensiveness** here clearly. According to Line 68-69, it seems the **comprehensiveness** just refers to different sizes of LLMs, one-turn prompt methods. However, this scope is insufficient for a text-to-SQL benchmark, lacking specific elements crucial to the field. so what is the point for this benchmark? Why do authors select text-to-SQL instead of other topics. I have some suggested aspects could be considered as comprehensiveness in text-to-SQL field:
        - Database schema distributions: classify and test LLM training in different database settings. Please note, with the same domain, same questions but **different database schema**, the generated SQLs and difficulties would be very different.
        - Question distributions: Analyzing performance across various question formats for the same database schema could also reveal some insights. The authors could reference Dr. SPIDER [3] as an example of this approach.
        - Training distributions: Training conditions should be explored since authors focus on fine-tuning mainly in this paper, including domain distributions, training data quantity, and data quality (clean vs. dirty). Compositional generalization data, such as SPIDER-SSP [11], would be also a good reference or example.
        - Authors just include few-shot or zero-shot one-pass prompting, ignoring more sophisticated but also popular settings such as Tree-of-Thought [4], Agenic workflow [5], also react-like prompting Inter-Coder [6]. Therefore, I think "comprehensiveness" is an over-sold word, potentially leading to reader disappointment after initial excitement.
        - Also authors should compare with other text-to-SQL benchmarks with comprehensive features to show its comprehensiveness by statistic tables. Just linguistic descriptions are not enough.
 - Regarding extensibility, the authors should clarify how new methods can be incorporated into the benchmark. If users need to design prompts and manually integrate multi-step or executed steps into the existing framework. The ease (or difficulty) of extending the benchmark to include these newer techniques should be discussed in detail. If the framework cannot readily support these newer approaches, or if significant human effort is required to integrate them, the authors should acknowledge these limitations. Moreover, if "extensibility" primarily refers to accommodating **new models** rather than **new techniques** specific to text-to-SQL tasks, this should be clearly stated, and the extensibility claims should be moderated accordingly.

8) The introduction is too short and very ambiguous. To improve clarity and impact, the authors could expand this section to include a more thorough discussion of their motivations and general approach, supported by illustrative figures or examples. An explicit discussion of Figure 1 within the introduction is necessary, highlighting the novel aspects of the proposed workflow. The introduction should provide a more comprehensive overview of related contexts, situating the current work within the broader field of text-to-SQL research.

After reading Introduction several times, I highly suggest that authors should reconsider the **unique** contributions of their benchmark compared to existing works. The introduction requires a more logical structure and careful use of terminology. Statements require to be moderated and supported by experimental or statistical evidence in subsequent sections. Each highlighted point in the introduction must have corresponding proof in the body of the paper. This approach will ensure that the true value and innovations of the benchmark in text-to-SQL research are accurately presented, setting appropriate reader expectations and emphasizing genuine contributions to the field.

## Task Definition:
1) The database D should be clarified as relational database. This paper doesn't consider other formats of databases like GraphDB or MongoDB. Whether database schema already contains db constraints such as PK or FK is a little controversial. In classic database textbooks [7][8], relational database schema **already contains** such constraints. Also, what is the reason authors try to decouple schema and constraints? I didn't see any discussions about distributions or features of such constraints to model performance. What does k mean (no description)? It seems that M just contain one triple of i-th exmaple?
2) In Line 99-100, k appears again, defined as the number of exemplars in few-shot prompting. It is unclear whether this k has the same meaning as the k defined in Section 2.1. If these are different concepts, using distinct variables would improve clarity. If they have the same meaning, authors should define this when it appears at the first time. Formula (1) presents a potential inconsistency. The authors define M as containing q,s,D, but in Formula (1), \sigma(q_i, M) implies \sigma(q_i, (q_i, s_i, D_i)) based on their definition. This raises the question of why q_i is included twice. Additionally, Formula (1) does not provide a clear definition of \theta and does not explicitly state what the **representation space** is, which is not intuitive. The distinction between large \theta and small \theta is also ambiguous since Formula 1 uses small \theta, while the interpretation mentions large \theta. These inconsistencies and absence of clear definitions make the formula and its interpretation difficult to comprehend.
3) In the \paragraph{Fine-Tuning}:
   - In Line 105-106, authors only include q and s as pairs for training. Why is the database not incorporated into the input, given that SQL queries can vary depending on the schema for the same question. The meaning of "serialized input of question" is unclear, because textual input should always be serialized tokens.
   - In Formula 2, what does \hat{s_i} mean? Does it mean gt SQL? if yes, why it models LLM_{theta}, or is \hat{s_i} the gt SQL?
   - Given that PEFT is the only training method tested by the authors, a more detailed definition by formulas would be beneficial. The current general formula does not effectively introduce how authors implement PEFT in such setting. The focus on LoRA, rather than other popular PEFT methods such as prefix-tuning [9] or adapter-tuning [10], seems weird with the claim of the authors of a "comprehensive" study. Including references for these alternative methods would strengthen the paper.

## Benchmark Construction:
The structure of this section should be polished and refined. Authors spent much space in introducing datasets, training, prompting from other works. Their own contributions of benchmarking pipleing by DB-GPT-Hub, just are described in 2 lines with Figure 1?  What are the unique aspects of this pipeline? How does this pipeline achieve comprehensiveness and extensiveness? Why are previous benchmarks insufficient for this task? What specific pain points does DB-GPT-Hub address? The are some suggestions and unambiguities in each part of this Section:
1) CodeBase: What does "suitable format" mean, one-sentence summarization should be clarified.
2) Metrics: The authors stated and realized the weakness of EM in Line 191-193. Given authors know this metric is not reliable, why did authors include it in experiments? Did authors mitigate the bias? How?
3) Base LLMs: Why didn't authors include more code llms such as Starcoder, Qwen-coder, Deepseek-coder since SQL is also one of popular programming language and they always present extraordinary performance in text-to-SQLs but general LLMs? This may also explain the fine-tuned code-llama had the best performance in this work.
4) Unclear Contributions: After reading this important section, I don't know what is the pain point of concurrent workflow, and what is special method / motivation in the benchmark construction or just unify other works? From my perspective, this work only combines data from other works keeping the same sqlite database, jsons, metadata, it seems not because of authors efforts. Also there are not new dimensions of evaluations, EX, EM, VES has already been utlized in many text-to-SQL works. Therefore, what is the critical contributions of  this work?

## Experiments:
1) The claim that CodeLlama "unsurprisingly" performs better requires more explanation. Why is it unsurprising? Is it due to SQL being a programming language and CodeLlama pre-training on related data? Or does logical reasoning from other coding languages transfer to SQL? To prove this, include more code-focused LLMs like Qwen-Coder, StarCoder, and DeepSeek-Coder for comparison. Dig deeper into what factors drive performance differences and what this means for text-to-SQL tasks. The analysis should go beyond surface-level observations to uncover meaningful insights about model capabilities and text-to-SQL characteristics.
2) Performance on Tuning: I think authors didn't even dig out codes or settings of two datasets. What can the observation "improvement on SPIDER is larger compared to BIRD" prove?  BIRD implements **hash set** comparison while SPIDER does not. Could this difference in evaluation methods affect the observed performance gains? Additionally, does the performance variation stem from domain differences between the datasets? A closer examination of result logs is necessary for meaningful insights. The term "complex" needs clarification. Does it refer to more complicated database schemas, lengthier questions, or a greater diversity of SQL keywords?
3) Fine-tuning vs Few-shot prompting: the decision to conduct experiments solely on SPIDER instead of both datasets raises concerns about potential bias in the conclusions. Why not include both SPIDER and BIRD to ensure more robust and generalizable findings. It is worth noting that the SPIDER leaderboard has not been updated since last year, rendering the current "second-best" open-source model rankings potentially outdated. For instance, DAIL-SQL, which may have been highly ranked on SPIDER, now only ranks **33rd** on the BIRD-SQL leaderboard. Many open-source models have surpassed it. This discrepancy highlights the importance of using up-to-date benchmarks and considering multiple datasets to draw accurate conclusions about model performance.
4) Fine-Tuning with More Exemplars:
  - First, what does **exemplar** mean in fine-tuning? I think it means in each data from trainning set, authors will give a fixed shots of examples as parts of demonstrations along with instructions, am I right? If so, please define this more clearly in Task Formulation, since it may lead to ambiguity that the number of exemplars refer to number of training data. If not, please tell the difference between exemplars and training data instances in fine-tuning.
  - In the previous analysis, authors used llama and qwen for one dataset. Now, reduced it further to qwen for one dataset. Why?
  - Why present EM here since authors already state EM is not reliable in Section about Metric? And performance gain distributions between EM and EX are not consistent, why is that?
  - What does random-shot mean here? Please use your formula to define this. Also, what is the random methods implemented exactly? For example, random 3 times, and report the average results or just randomly pick once which is highly biased? If you average several times, report div and mean with error bars please. Does your observation also happens in BIRD set? If not, why is that?

## Related Work:
some important works are missing: CoSQL, BIRD-SQL, Dr. SPIDER [3], SPIDER-SSP [11].

Overstatement: we admit it's hard and valuable to achieve a popular work in githubs. However, Github is not an academic place, the number of stars can not be used as the reason to illustrate its academic value since it has different evaluation aspects. A more academic way is to conduct a comprehensive user study to investigate why users like or star your benchmark more? For example, reasons behine higher stars are due to your better advertisement skills or some special components in your codebase that others do not have? The paper sumbission in ICLR is for research, can authors promise all users staring your project are doing research? All of them are active user account? Please give evidence if authors try to state anything in main content of paper.

Along this paper, I do not find anything special compared to SPIDER and BIRD. In BIRD benchmarks, there are works based on training (CodeS, granite, SuperSQL) and ICL works and agenic works. The contributions only are just combining several benchmarks into Githubs. In a short, I highly suggest authors consider an apprioprate way of highlighting their contributions in an academic way by more in-depth analysis, user study, and more taxonomies. Just summarizing and reproducing conlusions from cocurrent work are not qualified for researchers to accept it.

**References:** \
[1] LGESQL: Line Graph Enhanced Text-to-SQL Model with Mixed Local and Non-Local Relations, ACL 2022 \
[2] PICARD: Parsing Incrementally for Constrained Auto-Regressive Decoding from Language Models, EMNLP 2022 \
[3] Dr.Spider: A Diagnostic Evaluation Benchmark towards Text-to-SQL Robustness, ICLR 2022  \
[4] Tree of Thoughts: Deliberate Problem Solving with Large Language Models, NeurIPS 2023 \
[5] MAC-SQL: A Multi-Agent Collaborative Framework for Text-to-SQL \
[6] InterCode: Standardizing and Benchmarking Interactive Coding with Execution Feedback, NeurIPS 2023 \
[7] Database System Concept, 7th edition \
[8] Fundamentals of Database Systems, 7th edition, Page 160 \
[9] Prefix-Tuning: Optimizing Continuous Prompts for Generation, ACL 2021 \
[10] Adapter: Learning multiple visual domains with residual adapters, NeurIPS 2017 \
[11] Compositional Generalization and Natural Language Variation: Can a Semantic Parsing Approach Handle Both? ACL 2021

**Questions:**

See Weakness for details, I will stress on most crucial questions here, even if they are fundamental:

1) What is the pain point of previous text-to-SQL benchmarks, such as SPIDER and BIRD also support fine-tuning and there are training set and many works in the leaderboard already.
2) Why do you think such pain problems are important in text-to-SQL benchmark? And how do you resolve them?
3) How do you present your "comprehensiveness" and "extensiveness"? Show a table to compare against other works. Show how to compute "comprehensiveness", "extensibility". Given this paper even didn't include some popular techniques such as COT, TOT, React in prompt methods, DPO in training. Consider them as new techniques that users tried to extend in your workflow. What is the average human costs and efforts to extend them to show your extensibility. How about the previous work. Compare these to show your efficiency.
4) What does "examplar" mean in task formulation? Giving the first look at it mislead me to consider it as training data since authors use the letter i for all, which is confusing. Please define this clearly.
5) What is your benchmark workflow exactly? Just two lines are not clear. Is anything novel or special? Did authors resolve some issues of previous works which prevents them from popular github projects? If so what is the problem and how do Figure 1 resolve it?
6) What are new conclusions from experiments on your benchmark especially in terms of text-to-SQL aspects? Such as does domains / database schema/ training distribution of knowledge influence LLM performance? Which models in which settings are most robust?, which models are most potential? I encourage authors to read Dr. Spider [3] to learn how to conduct analysis and evaluate models in a more academic level. I highly encourage authors to learn how to bring out more insights from experiments based on your specific tasks. I think the mission of benchmark should bring out crucial research problems and reflect something interesting in this domain.

---

### Official Review · Reviewer_fE6Y · 2024-10-31

**Soundness:** 2
**Presentation:** 3
**Contribution:** 3
**Rating:** 3
**Confidence:** 4

**Summary:**

This paper focuses on generating database queries from natural language queries (text-to-SQL). Most recent research on text-to-SQL has concentrated on improving results by refining LLM prompts and optimizing interaction workflows with LLMs. The paper proposes a convenient tool, DB_GPT_Hub, which integrates data, model, and evaluation processes to form a complete and standardized code framework. However, the paper emphasizes experimental results related to fine-tuning methods more than a detailed introduction of the tool itself. Experiments were conducted using various sizes of Llama2, Qwen, and ChatGLM models, showcasing the effectiveness of fine-tuning methods. Additionally, a series of ablation studies were performed to discuss the impact of different settings during training.

**Strengths:**

1. The open-source framework proposed by the paper provides an integrated approach encompassing data construction, model training, and result evaluation. This robust framework is well-suited for research on text-to-SQL fine-tuning methods.
2. The experiments are thorough, examining something from sample sizes in training datasets to comparisons of LORA and QLORA fine-tuning methods and evaluations across different metrics, clearly identifying when fine-tuning proves effective.
3. The writing is clear and detailed, especially in defining the problem and describing the experimental setup.

**Weaknesses:**

1. The paper is more of an integration of the entire process for fine-tuning text-to-SQL tasks rather than a research study, making it somewhat lacking in innovation.
2. The paper offers limited detail on the framework’s structure. More granular information would enhance the paper’s completeness.
3. If the experiments included a comparison with results from prompt-based methods in closed-source LLMs, the paper could provide a more comprehensive analysis of the strengths and weaknesses of each approach.

**Questions:**

1. Why didn’t the authors use the more recent Qwen2 and Llama3 models? Could these newer models lead to different experimental outcomes due to their enhanced capabilities? Additionally, would a coder-specific model be better suited for the text-to-SQL task?

---

### Official Review · Reviewer_Zytt · 2024-11-04

**Soundness:** 3
**Presentation:** 3
**Contribution:** 3
**Rating:** 6
**Confidence:** 4

**Summary:**

The paper focuses on creating a benchmarking suite for LLMs in the area of Text to SQL. It introduces DB-GPT-Hub which is proposed to be an open source tool targeting the evaluation of LLMs mainly covering the fine-tuning use cases for text to SQL datasets/tasks. Since traditional fine-tuning is expensive for LLMs, this tool primarily covers PEFT such as LORA and QLORA. Their key contributions include a standard and comprehensive eval framework, modular and extensible codebase and the results of several open source models such as LLAMA, QWEN, BAICHUAN family of models and ChatGLM3 using their framework. The codebase supports many popular datasets such as SPIDER, BIRD, WikiSQL and others.

**Strengths:**

1. The paper's focus on creating a framework for evaluating fine-tuned LLMs is very relevant and has high impact potential as current benchmarks only support prompt based methods.
2. Proposed codebase is modular, supported evaluation metrics and fine-tuning methods are satisfactory and enables fair comparison for open sourced models. Additionally, supports closed models which is important.
3. Includes comprehensive analysis of LLMs including small and medium sizes while re-asserting the advantages of fine-tuning for performance critical and challenging tasks such as text to sql.
4. The experiments and results are well articulated and providing details related to GPU usage and time cost is a strength.
5. The paper is overall well detailed, readable from start to end, includes sufficient technical depth and is a valuable addition to the community.

**Weaknesses:**

1. It is not clear to the readers what are some existing tools that support prompt based evaluation and what are the strengths of the proposed tool compared to existing ones. The discussion on this is too short.
2. The paper's reported results on QLORA vs LORA performance for QWEN models seem unintuitive and probably require more discussion/insights into why the discrepancy compared to other models.
3. No discussion at all on ethical considerations. It is important to talk about how LLMs can hallucinate and generate incorrect answers which might lead to serious concerns for some industrial applications especially in health care, etc.
4. Limitations are not well-thought/rounded. It should include discussions related to limitations/biases of supported datasets, limitations of metrics as some LLMs might require post processing and the supported metrics might not be representative, etc.

**Questions:**

All the weaknesses could be corrected which can improve the overall strength and acceptance of the paper.

---

### Official Review · Reviewer_6957 · 2024-11-06

**Soundness:** 2
**Presentation:** 3
**Contribution:** 2
**Rating:** 3
**Confidence:** 5

**Summary:**

The paper proposes a framework for fine-tuning and benchmarking LLMs on popular Text-to-SQL datasets. The paper presents performance scores on the Spider and BIRD datasets, while the codebase supports other similar datasets, such as WikiSQL, CoSQL and Chase. The authors explore the performance of several open-weight LLMs on Spider and BIRD across different configurations (incl. base-model prompting, full-parameters fine-tuning or fine-tuning using LoRA and QLoRA adapters and training/testing with and without in-context demonstrations).

**Strengths:**

- The paper is generally easy-to-follow, and the experiments are clearly described.
- Experiments demonstrating the performance of popular LLMs (including models consisting of up to 72B parameters) across different configurations.
- The provided codebase could be of some value for exploring the performance of fine-tuning-based baselines on the involved benchmark datasets.

**Weaknesses:**

- Nowadays, state-of-the-art performance in NL2SQL is achieved by systems that extend beyond simple fine-tuning of an autoregressive LLM on (question, SQL) pairs. These systems typically incorporate features such as, schema and value linking, SQL validity checks, and multi-turn reflection and refinement. However, the proposed benchmarking framework focuses on exploring the performance of *more baseline* solutions.
- The current approach does not account for unique characteristics or idiosyncrasies present across the various supported datasets. For instance, the BIRD dataset, includes an `evidence` field for each question, and, in many cases this information is necessary, for coming up with the appropriate SQL for each input. However, the paper does not include any considerations for such cases. This may actually explain why the execution accuracy scores provided in Table 2 are lower compared to the ones reported on the official leaderboard page of BIRD (i.e. https://bird-bench.github.io).
- A significant number of systems published in the corresponding leaderboards leverage the DB content for each input DB (Wang et al., 2020; Shen et al., 2024; Sun et al., 2024). However, my understanding is that this is not currently supported by the proposed framework.

Zhili Shen, Pavlos Vougiouklis, Chenxin Diao, Kaustubh Vyas, Yuanyi Ji, and Jeff Z. Pan. 2024. Improving Retrieval-augmented Text-to-SQL with AST-based Ranking and Schema Pruning. Preprint, arXiv:2407.03227.

Ruoxi Sun, Sercan Ö. Arik, Alex Muzio, Lesly Miculicich, Satya Gundabathula, Pengcheng Yin, Hanjun Dai, Hootan Nakhost, Rajarishi Sinha, Zifeng Wang, and Tomas Pfister. 2024. SQL-PaLM: Improved Large Language Model Adaptation for Text-to-SQL (extended). Preprint, arXiv:2306.00739.

Bailin Wang, Richard Shin, Xiaodong Liu, Oleksandr Polozov, and Matthew Richardson. 2020. RAT-SQL: Relation-aware schema encoding and linking for text-to-SQL parsers. In Proceedings of the 58th Annual Meeting of the Association for Computational Linguistics, pages 7567–7578, Online. Association for Computational Linguistics.

**Questions:**

- I believe the codebase could benefit from additional layers of abstraction to enhance experiment configuration, such as wrappers for adjusting the format of in-context demonstrations or the format of the DB schema used for training and test data instances.

---

### Note · Authors · 2024-11-13

I have read and agree with the venue's withdrawal policy on behalf of myself and my co-authors.